# Cognitive Neuroscience Methods in Enhancing Health Literacy

**DOI:** 10.3390/ijerph18105331

**Published:** 2021-05-17

**Authors:** Mateusz Piwowarski, Katarzyna Gadomska-Lila, Kesra Nermend

**Affiliations:** 1Department of Decision Support Methods and Cognitive Neuroscience, University of Szczecin, 71-004 Szczecin, Poland; kesra.nermend@usz.edu.pl; 2Department of Organization and Management, University of Szczecin, 71-004 Szczecin, Poland; katarzyna.gadomska-lila@usz.edu.pl

**Keywords:** public health, health literacy, cognitive neuroscience, EEG, advertisement

## Abstract

The aim of the article is to identify the usefulness of cognitive neuroscience methods in assessing the effectiveness of social advertising and constructing messages referring to the generally understood health promotion, which is to contribute to the development of health awareness, and hence to health literacy. The presented research has also proven useful in the field of managing the processes that improve the communication between the organization and its environment. The researchers experimentally applied cognitive neuroscience methods, mainly EEG measurements, including a metric which is one of the most frequently used to measure the reception of advertising messages, i.e., frontal asymmetry. The purpose of the study was to test cognitive responses as expressed by neural indices (memorization, interest) to the reception of an advertisement for the construction of a hospice for adults. For comparative purposes, a questionnaire survey was also conducted. The research findings have confirmed that there are significant differences in remembering the advertisement in question by different groups of recipients (women/men). They also indicate a different level of interest in the advertisement, which may result from different preferences of the recipients concerning the nature of ads. The obtained results contribute to a better understanding of how to design advertising messages concerning health, so that they increase the awareness of the recipients’ responsibility for their own health and induce specific behavior patterns aimed at supporting health-related initiatives, e.g., donating funds for building hospices or performing preventive tests. In this respect, the study findings help improve the organizations’ communication with their environment, thus enhancing their performance. The study has also confirmed the potential and innovativeness of cognitive neuroscience methods as well as their considerable possibilities for application in this field.

## 1. Introduction

A comprehensive approach to health requires constant strengthening of public health awareness, to which health literacy refers. It is also essential for health promotion, especially with regard to preventive health issues, as well as to people’s involvement in various initiatives, such as supporting health promotion campaigns. In a narrow sense, health literacy is the ability of individuals to seek, process, and understand information necessary to make appropriate health decisions [1], the ability to make health-related decisions in the context of everyday life, including at home, in the community, in the workplace, and in the health care system [2]. The term also refers to a specific skill that is needed to successfully manage a large number of health-relevant tasks and decisions to be made each day [3,4]. Health literacy encompasses those cognitive mechanisms and social skills that affect the motivation and ability of individuals to successfully obtain, process, and use information in order to maintain or improve their health [5], such as reading, writing, counting, and retrieving information [6], using multimedia technologies, or problem solving [7,8]. However, health literacy depends on many factors. The determinants of individual health awareness are demographic and socio-cultural factors such as occupation, income level, social status, culture or language, as well as more personal factors, such as individual experience of illness and contacts with the health care system, or individual traits such as age, gender, race, education level, etc. [9]. These factors should be taken into account when selecting instruments for enhancing health awareness and health promotion as they influence the effectiveness of the steps taken.

A broader view of health literacy is related to empowerment. It emphasizes social skills, mainly in the area of communication or negotiation, that are necessary in the process of making health promoting decisions in practice [10]. From a public health perspective, the above is crucial in terms of improving health of whole communities, not just individuals [11]. This means that an inalienable part of health literacy is education [12]. Health literacy is also a useful tool in identifying health information delivery issues and empowering people to use this information to make proper health decisions [13]. From a public health perspective, what seems to be of particular importance is civic literacy [14], which relates to involvement in public life with its decision-making processes. The term refers to the skills that help a person (citizen) to be aware of public affairs and to be involved in decision-making processes. The key skills in this respect are: conscious and critical perception of media and their rational use as a tool for work, communication and learning (media literacy), as well as knowledge about social and governmental processes or the awareness that personal health-related decisions can affect the health of the whole community [14]. The idea underlying this change is that both public and private organizations should be considerably involved in efforts to improve health care systems and the health of individual citizens. In this view, health literacy points to a health care organization that uses strategies that make it easier for patients to engage in the process, navigate through the health care system, understand health information, and last but not least manage their health [15,16]. This highlights the importance of specific attributes of a health literate organization. The key ones include promotion of health literacy and its effective communication, which is also seen as an important attribute of a health literate organization. Communication is one of the key factors influencing the success of an organization. The recognition of needs, the choice of messages sent and their effectiveness are important for both the efficient implementation of management processes in the organization and for its contacts with the environment [17,18]. This requires effective tools to deliver the right information and provide the support people need to use the information when making correct health decisions [19]. Consequently, a need arises to find effective methods and tools to create health promoting messages, e.g., advertisements. Organizations operating within the health protection and care system often use advertising messages to achieve specific objectives related to their business profile. This group of organization includes hospices. Being institutions operating within the health care system, hospices often turn to the advertising messages to raise funds for their operation. The addressees of their messages may be both a wide range of recipients as well as a narrow group with a particular profile, hence the question about the effectiveness of such a message.

The purpose of this paper is to identify the usefulness of the methods of cognitive neuroscience when testing the effectiveness of public service advertising and designing messages concerning generally understood health promotion, which is going to contribute to public health awareness, to which health literacy refers.

Most often the evaluation of the advertising message is based on the opinions of its potential viewers. The research methods commonly used to assess the quality of such advertisements include focus groups, individual in-depth interviews, or population surveys (e.g., online, telephone) [20,21,22,23]. However, despite the advantages, the above research techniques also have certain limitations. The respondents’ answers may depend on the interaction context, e.g., on the way the questions are phrased, the answers given by other respondents or by the focus group leader. Therefore, in many situations the effect of the advertisement on viewers is difficult, or even impossible, to assess with verbal measures [24,25]. Sometimes people sense that they know about something, but they are not able to extract this knowledge from their memory and transfer it in a verbal form. This can be explained by the functioning of their subconsciousness where a variety of experiences from every moment of a person’s life is stored. It is assumed that up to 95% of thoughts, emotions, learning processes, and memorizing occurs beyond our consciousness. Research has shown that the majority of events reach consciousness only after about 300 ms from the occurrence of the stimulus. This means that shorter interactions, although recorded by the brain, are not consciously perceived and thus cannot be subject to verbal judgments [26]. The processes occurring in the subconscious, despite not being consciously controlled by a person, have a significant effect on their functioning and on the decisions they make [27,28,29]. In order to better understand some psychological processes, it is therefore advisable to analyze the reactions of the human brain and body to external stimuli perceived [30,31]. The study of physiological (including neurophysiological) parameters of the human body which determine the response to the stimuli in question can significantly expand the range of information about the reception of a given message. It allows researchers to obtain information unavailable for declarative research. Scientific research shows that males and females differ in brain activation during cognitive tasks. Research concerns among other things, tasks of mental rotation, visual stimulation, emotional recognition, working memory and verbal processing [32,33,34]. This also applies to the perception of TV advertising [35,36,37,38,39]. Therefore, in this article we focus on the analysis of the perception of advertising based on sex. Due to sexual dimorphism of the human brain, this type of research should be designed accordingly. The gender-related differences do not arise in every situation but, if they do, they can provide valuable guidance in designing announcements and advertising messages. The potential that lies in the methods of cognitive neuroscience which encourage their use in the area of building health awareness. The application of specific techniques helps design the message in such a way that it will reach a specific group of addressees more effectively. This can help shape the health awareness of a given group of recipients in the most desirable way, thus increasing the effectiveness of the organization’s activities. Therefore, it seems justified to consider the use of cognitive neuroscience methods in this field to be innovative.

## 2. Cognitive Neuroscience Methods in the Study on Advertising Messages

The dynamic development of medical, biological, chemical, computer, or psychological sciences has led to the emergence of a new area of scientific research, i.e., cognitive neuroscience which integrates many scientific issues, such as psychology of cognitive processes, anatomy or human physiology (including human brain). It focuses on the neural basis of mental processes, combining theories of cognitive psychology and computer modeling with experimental data on the brain [40,41]. Methods of cognitive neuroscience supported by other techniques for measuring human physiological processes more and more often are finding applications in the studies on the multimedia content, such as video advertisements. EEG (electroencephalography) measurements, fMRI (functional magnetic resonance imaging), as well as GSR (galvanic skin response), HR (heart rate), EMG (electromyography) and eye tracking are used for this purpose [35,36,42,43,44,45,46,47,48,49,50,51,52,53,54,55,56,57,58,59,60]. Measurement techniques based on neuroscience are less prone to assessment errors and are more objective. They allow information to be obtained at its source (in the brain) before it is presented in the form of an opinion or judgment (e.g., surveys or interviews) [61,62,63,64]. Some of them, EEG for instance, are characterized by a very high temporal resolution, which allows for time-accurate measurements of people’s reactions to the presented stimuli [43,48,65]. Various EEG metrics based on brain activity in its different parts and at different frequency ranges are used to assess the reception of media messages (advertisements). They are utilized, among others, to study the reception and processing of video stimuli. Numerous studies have confirmed that the effectiveness of advertising depends on cognitive processes of attention, memory and emotions [36,44,47,55,65,66,67,68,69].

Frontal asymmetry is one of the EEG metrics that are best described in the literature and most frequently applied in assessing the receipt of advertising messages. It is recorded at rest and under task conditions (when responding to a stimulus). The frequency range closely associated with frontal asymmetry is primarily the alpha band (8–13 Hz), but theta (4–8 Hz), beta (13–30 Hz), and even gamma (above 30 Hz) bands are also used [70,71,72]. Frontal asymmetry is an indicator of asymmetrical brain activity in the frontal cortex between the left and right hemispheres. It is associated with emotional states and motivational systems that translate into human behavior. According to Davidson’s model, left frontal brain activity indicates a tendency to approach a stimulus, to become interested in the stimulus. Relatively greater right frontal brain activity is associated with negative occurrences, withdrawal, and stimulus avoidance [52,73,74,75]. Stronger lateralization of frontal alpha (left–right) has been observed in a number of studies on advertising messages, in response to the positively valenced content of video scenes and advertising clips [21,36,53,76]. It has been linked to dopaminergic reward system activation [76,77]. Frontal asymmetry alpha is considered as a biomarker of the behavioral activation system [67,78]. Based on this knowledge, an Approach Withdrawal index can be defined to reflect approach or withdrawal behavior in relation to the presented stimulus.

The measurements of the electrical activity of the frontocentral parts of the brain permit study of the processes of encoding and memory retrieval. An increase in the power of the EEG signal in this area within the theta range (4–8 Hz) and at higher gamma frequencies is associated with the encoding process [79,80,81]. According to the HERA hemispheric encoding/retrieval asymmetry model, the left part of the prefrontal cortex is more involved in the encoding process of episodic memory. In contrast, the right prefrontal cortex is responsible for retrieving this memory [82,83,84]. The assumptions for this model are consistent with the results of many other studies which have confirmed an increase in theta and gamma power in the frontal parts of the brain. This applies to both memory encoding processes and to the memorization of different types of stimuli [80,85,86,87,88]. These properties have been utilized in a variety of studies on remembering television messages, advertising clips, and the impact of video stimuli [21,54,89,90,91,92,93].

The analysis of the world literature on the applicability of EEG signal measurements in the reception of advertising messages, as well as our own previous studies [56,57,58,60,94,95,96,97,98] confirm the correctness of the adopted research direction. Reaching and analyzing the brain response to the specific stimuli (in this case, scenes shown in public service advertisements) may bring measurable benefits to the entire health care system. Designing effective messages enhancing health literacy and those emphasizing the need to help people in difficult health situations may significantly support (financially, organizationally or image-wise) the functioning of organization dealing with broadly understood human health. Hence, the idea appeared of a research project where measurements of neurophysiological parameters are utilized and the thesis about the effectiveness of this type of research is verified.

## 3. Materials and Methods

### 3.1. Participants

The cohort consisted of thirty-one subjects (16 females and 15 males) with an age range between 22 and 68 years. The mean age and standard deviation for all participants were: 42.09 (SD = 3.83); for women: 40.75 (SD = 11.59); for men: 43.53 (SD = 6.39). The study subjects were university students as well as employees working in a variety of occupations. There was only one left-handed person in the research group, which is about 3% of the total (this aspect was not considered in EEG analysis). They were healthy individuals who voluntarily submitted to the research procedure. The experiment was conducted in compliance with the Declaration of Helsinki (as updated in 2013) [99] and was approved by the Bioethics Committees at Regional Chamber of Physicians in Szczecin. Prior to the start of the research project, participants were informed about the study protocol (without providing details), as well as about the measuring devices used. All the participants gave their written consent to be included in the study and to their personal data processing.

### 3.2. Protocol and Stimuli

The study was intended to reflect a real-life television viewing situation. The film shown was separated by two blocks of commercials. The film was a documentary about the natural environment and the advertisements were of different nature, both commercials and public service announcements. There were 6 video ads in the first advertising break and 7 in the second break, and they were randomly placed. The duration of the ads varied from 15 to 30 to 60 s. The advertisement in question was released in the second block and lasted for 60 s. The commercial was titled “Loved ones pass away happy” (vimeo.com/309422044 accessed on 21 September 2020) and was meant to encourage people to financially support the construction of a hospice (Figure 1).

The storyline of the public service advertisement takes us emotionally from a state of sadness (an old man lying on a hospital bed) to the realm of dreams (a rocket trip into space). In the ad there are scenes showing: the figure (face) of an elderly man and a child, their walk together through a desert to the rocket, the interior of the rocket, their figures in spacesuits on seats, and finally an announcement is displayed encouraging people to support the construction of a hospice.

For the purposes of the study, seven specific scenes were extracted from the 60-s commercial. Each of them presents thematic fragments that are interesting in terms of their reception by the surveyed respondents. Scenes were extracted with an accuracy of 0.5 s (in brackets time intervals in seconds):(S1) An elderly (sad) man lying on a hospital bed (0 ÷ 6.5);(S2) The smiling man wearing a spacesuit (12.0 ÷ 13.0);(S3) A child reaching out to the man and then walking with him toward a rocket (13.5 ÷ 25.0);(S4) The man with the child walking inside the rocket (25.5 ÷ 31.0);(S5) The man fastening his seatbelt while seated with the child in the rocket seats (31.5 ÷ 40.5);(S6) The smiling man wearing a buckled astronaut helmet (41.0 ÷ 44.0);(S7) Announcement (text and voiceover appearing) about building a hospice and asking for financial support (48.0 ÷ 60.0).

The entire advert was in a melancholic mood. The video was accompanied by soothing music. Voiceover appears only in scene S7.

The study was conducted in two stages. First, the participant’s brain activity and, additionally, their electrodermal activity and heart rate were recorded. The initial phase of the measurement procedure involved the identification of a reference point (baseline), relative to which the neurophysiological responses of the subjects were determined. The stimulus used to determine the baseline was a photograph of a neutral nature (in line with IAPS [100]). The participants were asked to stare at the image presented on the monitor screen for a period of 1 min. In the second part of the study, the participants completed a survey questionnaire in a separate room. The questionnaire was used to check whether the participants had not seen the advertisement before, whether they memorized anything, and to learn their opinion about the advertisement in question (according to different criteria). Following the subsequent steps of the research protocol, the respondents were asked questions by the interviewer. Initially, he asked which adverts they had memorized (in that short period of time). Then, he asked for a detailed description of the storyline of the advertisement. The details provided by the respondents were written down by the interviewer. Then, after the ads had been shown again, the respondents were asked to rate them.

### 3.3. Apparatus and EEG Recordings

The electrical activity of the subjects’ brains was recorded using a g.Nautilus mobile device from g.tec. What was interesting for the study was the frontal lobe. A total of 8 measuring (wet) electrodes were placed in the locations: Fp1, Fp2, F3, F7, Fz, F4, F8 and Cz, following the 10–20 standard. The ground (GND) was placed on the skin of the head, while the reference was placed on the left earlobe. Caps used in the study: g.GAMMAcap^2^, size: medium and large. Measurements were performed at an electrode–skin impedance below 10 kΩ. The sampling frequency of the EEG signal was 500 Hz, noise level: <10.6 µV RMS between 1 and 30 Hz (at highest input sensitivity).

The EEG signal was processed and analyzed using the Matlab package with EEGLAB toolboxes, FieldTrip and the authors’ innovative software. EEG preprocessing included several steps [101]. In the first stage, frequencies of no interest due to the nature of the study were filtered out. The fifth-order Butterworth filter [2–30 Hz] was used to eliminate the slow-variable trend and high-frequency interference of, e.g., muscle origin. The electrodes were then placed and events were imported (identification of individual advertisements). Any bad channels they were removed (max. one channel) and then interpolated. Different approaches were used to eliminate artifacts from the EEG signal. Interference from electrodes was flagged and excluded from further analysis. Ocular artifacts were extracted using the ICA method (Infomax algorithm) [102]. Only one of the identified components was removed. After the signal was reconstructed, the process of eliminating epochs containing other artifacts was performed. For this purpose, the continuous signal was divided into one-second time windows and interference analysis was performed in such intervals. The following criteria for artifact determination were adopted: amplitude exceeding V and slope of the trend between windows exceeding V/s, with R-squared = 0.3. Epochs marked as artifacts were removed from further analyses. A cleaned EEG signal was thus obtained and subjected to further processing and analysis.

### 3.4. Indicators of Advertising Receipt Evaluation

An Individual Alpha Frequency (IAF) was determined for each participant. The alpha and theta frequency bands analyzed in the study were determined based on the IAF. That was done according to the notation IAF ± x, where x is the integer used to define the beginning and end of the band in Hz. The alpha band was defined as (IAF-2Hz ÷ IAF + 2 Hz) and the theta band was defined as (IAF-6 ÷ IAF-2) [86,103]. Global field power (GFP) was calculated for the obtained bands. GFP is a quantity that describing the total EEG activity at specific measurement points at a given time [104,105]. It corresponds to the standard deviation of the mean EEG amplitude from the electrodes at a given moment. Additionally, it is a parameter for analyzing EEG in the function of time. GFP was calculated from a specific set of electrodes according to the formula [53]:(1)$$GFP=\frac{1}{N}\overset{N}{\underset{i=1}{\sum }}x_{\vartheta_{i}}{\left(t\right)}^{2}$$
where: *ϑ* is the considered EEG band,
*N* is the number of electrodes included in the area of interest,*i* is the electrodes index,*x* is the EEG sample for time *t* filtered for a given bandwidth *ϑ* and for a given channel *i*.

From the GFP values, indices were calculated to determine memory processes (MI index) and the degree of interest in the presented stimuli (AW index).

To determine the MI index, data obtained from EEG measurements, from electrodes located above the frontal left part of the brain (Fp1, F3, F7), were used. The frequency band considered was theta. The MI index was calculated according to the following formula [54,83,84]:(2)$$MI=\frac{1}{N_{Q}}\underset{i\in Q}{\sum }x_{\theta_{i}\left(t\right)}^{2}=AveragePower_{\theta_{left\;frontal}}$$
where: $x_{\theta_{i}}$ represents the *i*-th EEG channel in the theta band,

Q is the set of left channels (Fp1, F3, F7),$N_{Q}$ represents its cardinality.

Higher MI values are interpreted as better memorization at a given point in time.

The AW index was determined on the basis of EEG data from measuring electrodes located in the frontal part of the brain, in both hemispheres (left side: Fp1, F3, F7, right side: Fp2, F4, F8). The frequency band included in the calculations covered the alpha waveband. The formula by which the AW index was calculated is as follows [52,75,106]:(3)$$AW=\frac{1}{N_{P}}\underset{i\in P}{\sum }x_{\alpha_{i}}^{2}\left(t\right)-\frac{1}{N_{Q}}\underset{i\in Q}{\sum }y_{\alpha_{i}}^{2}\left(t\right)=AveragePower_{\alpha_{right\;frontal}}-AveragePower_{\alpha_{left\;frontal}}$$
where: $x_{\alpha_{i}}$ and $y_{\alpha_{i}}$ represent the *i*-th EEG channel in the alpha band,
P and Q are the sets of right channels and left channels,$N_{P}$ and $N_{Q}$ represent their cardinality.

The approach/withdrawal (AW) index has been defined as the frontal alpha asymmetry. It is interpreted as a motivation to approach a stimulus or withdrawal. Higher values of it mean an increase in interest (approach) in the presented stimulus (advertising scenes), and lower values mean a decrease in interest (withdrawal) [73].

Eventually, the MI, AW index values were normalized (z-score) for each second of the advertisement using the mean and standard deviation of the index values calculated for the baseline. These calculations were performed as follows [35]:(4)$$Normalized\;index\;\left(z-score\right)=\frac{index\;value-mean\;value\;\left(baseline\right)}{standard\;deviation\;\left(baseline\right)}$$

### 3.5. Statistical Analysis

A two-way analysis of variance (ANOVA) was used for the statistical analysis of the obtained EEG test results (MI, AW indices). The dependent variables were the values of MI and AW indices. Averaged values were used over three channels (left side: Fp1, F3, F7, right side: Fp2, F4, F8) for each participant. The first factor (independent variable) was the GENDER of the study participants considered at two levels: men (M) and women (W). The second factor was the SCENES of the advertisement analyzed at seven levels: scene 1 (S1), scene 2 (S2) and so on to scene 7 (S7). The assumptions of analysis of variance, namely tests for normal distribution of the dependent variable scores across groups, and homogeneity of variance were tested. Both main effects of factors (GENDER, SCENES) and the interaction effect (GENDER x SCENES) were analyzed. The number of repetitions of each combination GENDER x SCENES was 1 and the number of all experimental units was 14. What was examined was the interaction of these two factors on the values of the calculated indices. Having established the fact of the overall F-test significance (the analysis of variance), post hoc comparisons were performed. They permitted assessment of between which groups statistically significant differences had occurred. The analyses were expected to provide answers to the following research questions:Are there any differences in the mean values of MI, AW by gender?Are there any differences in the mean values of MI, AW for different advertising scenes?Are there any interactions between the GENDER and SCENES factors?Between which groups of the SCENES factor are there differences as regards the mean values for the GENDER factor groups?

The significance of the relationship between the answers of men and women to the survey questions was also tested. For this purpose, the chi-square test of concordance was used ($\chi^{2}$).

## 4. Research Results

### 4.1. Memorization Index

When examining the effect of the GENDER factor, it was found that there were significant differences in the mean MI values between men and women: F (1, 2127) = 51.92, *p* < 0.005, $\eta_{P}^{2}=0.024$. Women obtained higher memorization values (M = 0.34, SE = 0.03) than the male group (M = −0.04, SE = 0.04). No statistical significance was found as regards the effect of the SCENES factor: F (6, 2127) = 0.97, *p* = 0.442, $\eta_{P}^{2}=0.003$. Similarly, there was no statistical significance as far as the “Gender x Scenes” interaction effect was concerned: F (6, 2127) = 0.64, *p* = 0.132, $\eta_{P}^{2}=0.005$.

When comparing the distribution of mean MI values (95% CI) recorded for men and women, a different distribution can be seen for each of the scenes under study (S1–S7) (Figure 2).

Women obtained higher mean MI values for all scenes than the male group. Particularly in the case of S2, the mean MI values were high (M = 0.536), while for S6 they were slightly lower (M = 0.449). The lowest mean values in the group of women were recorded for S3 (M = 0.24), S4 (M = 0.239) and S7 (M = 0.222). In the group of men, the mean MI values were the highest for S5 (M = 0.093), S3 (M = 0.043), and S1 (M = 0.039), respectively, and the lowest for S2 (M = −0.367).

In order to find out for which scenes the differences in mean values between men and women were the biggest, a pairwise comparison was made. That comparison was intended to provide the answer whether the content of the scenes (a sad man laying in bed, a smiling man, a child with a man, etc.) affected the differentiated levels of memorization by gender. The applied Tukey test (confirmed by the Bonferroni test) indicated a significant difference in MI values between the men and women as regards S2 (*p* = 0.014). In the case of other scenes these differences were irrelevant (S1: *p* = 0.295, S3: *p* = 0.606, S4: *p* = 0.214, S5: *p* = 0.421, S6: *p* = 0.184, S7: *p* = 0.379).

### 4.2. Approach–Withdrawal Index

The analyses of Approach–Withdrawal index values revealed no significant effect of the GENDER factor: F (1, 2129) = 0.14, *p* = 0.70, $\eta_{P}^{2}=0.00007$. A statistically insignificant result was also obtained regarding the SCENES factor: F (6, 2129) = 0.87, *p* = 0.52, $\eta_{P}^{2}=0.0024$. For the interaction effect of both “Gender x Scenes” factors, significant differences were found between the mean AW index values for men and women: F (6, 2127) = 2.24, *p* = 0.037, $\eta_{P}^{2}=0.063$.

The distribution of mean AW values (95% CI) for the male and female groups, for each scene (S1–S7) is shown in Figure 3.

The highest mean AW values were obtained by women for S2 (M = 0.11) and S6 (M = 0.08), and the lowest for S7 (M = −0.23). As regards male participants, the scene with the highest mean AW index value was S7 (M = 0.09), while for S1 the index was the lowest (M = −0.23). Post hoc analysis (Fisher F-test) showed that the only statistically significant difference for this interaction effect was obtained when comparing the mean index values of GENDER factor regarding S7. Here, the male participants (M = 0.09, SE = 0.07) achieved higher AW values than the female group (M = −0.23, SE = 0.07). For the other scenes, the comparisons of the mean index values proved to be statistically insignificant (S1: *p* = 0.0318, S2: *p* = 0.459, S3: *p* = 0.386, S4: *p* = 0.918, S5: *p* = 0.812, S6: *p* = 0.286).

Scenes S2 (MI index) and S7 (AW index) appeared to be of key importance in terms of differences in the advertisement receipt between men and women. Scene S2 depicted a smiling elderly man dressed in a spacesuit. His smile activated stronger brain processes responsible for memorizing and interest in women than in men. In the group of women, that scene evoked the highest MI and AW index values. Another scene in which the man’s smile appears is S6 (a man wearing a space helmet). Similarly to S2, the MI and AW index values for S6 amount to high values in women (in the group of men the indices are lower than for most other scenes).

The frames extracted from the advertisement video for S2 (at 0.5 s intervals) are shown in Figure 4a, while for S6 (at 1 s intervals)—in Figure 4b.

As can be seen in Figure 4, the film shots in these parts of the advertisement are directly focused on the face, and the man’s smile is very subtle. In S2, the change in his facial expression (smile) continues for a total of about 1.5 s, and in S6 it lasts slightly longer, for about 4 s. However, this was sufficient to elicit brain responses translating into varying receipt of the scenes by gender.

S7 divided into frames (at 2 s intervals) is shown in Figure 5.

This is the final advertisement part which continues for 12 s and contains information encouraging people to support the construction of an adult hospice. The scene is monochrome (black and white). The visual part is complemented by a voiceover that appeals for financial support via the provided website address.

Measurements of the electrical activity of the brain have shown that the high EEG temporal resolution makes it possible to examine even the elusive fragments of the message that last fractions of a second. Many of such stimuli remain beyond the conscience of the recipient. Therefore, it seems almost impossible to express any opinion about such fragments.

### 4.3. Survey Results

One of the research objectives was also to compare the information gained from the neurophysiological measurements with the declarative information acquired from the questionnaire. Following the EEG measurements, respondents were asked if they remembered seeing the hospice advertisement shown during the study session (video interrupted by commercial breaks). More specifically, the question was worded as follows: Advertisements of which brands, products, services did you just see while watching the film? Various names were then mentioned by the interviewer, both of brands, products, services whose advertisements were shown and of those which were not (24 in total). When asked this question and the name of the hospice mentioned, a positive response was given by 74.19% of the respondents: 23 subjects, 14 men (about 61%) and 9 women (about 39%). The value of the chi-square statistic was shown to be statistically significant: $\chi^{2}\left(1,\;\;N=31\right)=5.560;\;p=0.018$. This means that there were relevant differences between women and men in terms of remembering the advertisement in question (with so formulated question).

When asking the respondents about the details remembered from the advertisement, i.e., what it was about, what was its main idea and what it advertised, in most cases the respondents had trouble describing the scenes they had seen. About 45% of the respondents (including 40% men and 60% women) were able to say something about this advertisement, most of them in very general terms. As regards scenes S2 and S6, only 6.45% of the respondents remembered a smiling and happy elderly man (including 50% men and 50% women). In contrast, the message contained in scene S7 was remembered (at least partly) by 16.13% of respondents (including 40% men and 60% women).

The survey was continued after the advert was shown again. The respondents were asked if they generally liked the commercial, if the message was clear, if it inspired them to donate to support the hospice construction, and if they thought it might encourage other people to donate to that cause. The answers were given on a four-point scale: 0—definitely NO, 1—rather NO, 2—rather YES, 3—definitely YES. They were also asked whether they would like to share any comments on the advertisement. Their answers were analyzed with a particular emphasis on the gender aspect. The results, along with an analysis of statistical significance, are shown in Table 1.

Of the four survey questions posed, a statistically significant difference between men’s and women’s responses was found only regarding the question of whether the advertisement was liked. The male respondents assessed it much more positively. The remaining answers concerning the advertisement evaluation did not differ between men and women. The remaining answers concerning the advertisement evaluation did not statistically differ between men and women, though the male respondents assessed it slightly more positively.

## 5. Discussion

The purpose of this study was to find out whether the brain activity (in its frontal part) evaluated by changes in the EEG signal power in the theta and alpha bands can be used to assess and explain the reception of advertising. According to current knowledge and proven research results, a change in theta power in left frontal region is associated with memory processing [21,43,47,55,107]. Increased cortical activity in this area during advert viewing is interpreted as a memory (encoding) process. This is consistent with the HERA model which assumes that the left prefrontal cortex (PFC) is more involved than the right PFC in encoding episodic memory. In turn, the right PFC is more involved than the left PFC in episodic memory retrieval [82,83,84]. By obtaining higher MI values in females for scenes S2 and S6 (based on the GFP power of the EEG signal in the theta band), we obtain confirmation of the increased level of encoding of these scenes in episodic memory. The influence of observed emotions (positive and negative) on the memory process is confirmed by numerous studies [108,109,110,111]. This fact can be explained by the higher MI values in women for scenes S2 and S6. In the context of the whole advertisement which has a melancholic, sad character, the scenes with positive emotions (smile) may gain special importance. However, the situation is different in men (weaker memory encoding). These differences can be partially explained by different preferences regarding the nature of the adverts. Women have a greater preference for adverts that are active but pleasurable to watch, whereas men prefer them to be active but less pleasurable [36]. However, taking the specific example of the effect of emotional scenes (positive, smile) on men’s and women’s receipt of this type of advertising, the results are not conclusive. This issue requires further examination.

On the other hand, differences in signal power changes in the alpha frequency range between the right and left frontal regions of the brain (frontal asymmetry) allow us to examine the degree of interest in the presented stimulus (the advertisement scenes). This is consistent with the assumptions of Davidson’s approach/withdrawal model [21,112,113,114]. Higher values of frontal alpha asymmetry reflect greater left hemisphere activity. Note that alpha power is inversely related to cortical activity, and the decreased values of alpha band power indicate increased cortical activity (higher activity—disappearance of the alpha rhythm). What is more, using the logarithm when determining the difference in power makes it possible to provide a correction for the overall alpha power. This is of practical importance, if only because of individual structure (thickness) of the human skull, which in turn affects the signal amplitude [72,75,115]. Stimulating the respondents with visual and auditory material (advertisement) allowed us to determine approach and withdrawal patterns in reaction to the stimulus. The higher women’s AW values regarding scenes S2 and S6 were attributed to the positive reception of these particular fragments. The protagonist’s smile thus acted similarly to memory encoding. Significant differences in interest (represented by AW) between men and women were seen regarding S7. Relative to the other scenes, S7 visually did not appear to be attractive (Figure 5). The white background transitioning to black with information about the construction of an adult hospice produces a saddening effect. The introduction of an additional stimulus in this scene, namely the voiceover, was probably supposed to emphasize the purpose of this advertisement. This type of stimuli (sound effects) enhances the dynamic change of affective states, which is reflected in the brain processes [106]. The power of these changes and its impact on the viewer, including various emotional states, have been confirmed by numerous studies [116,117,118]. However, it is important to note that the audio impact is determined by the viewers’ individual preferences correlated with their gender, age, personality or cognitive style [106]. The voiceover narration limited to S7 only was designed to capture the attention and evoke specific emotional states in the audience. The results of the present study suggest that such a combination of message elements as used in scene S7 (black background, white subtitle, voiceover replacing the background music of the advertisement) may be more effective in arousing interest in men, while in women the opposite is true.

The comparison of electroencephalographic measurements with behavioral data (questionnaire) on remembering revealed significant discrepancies. In the case of the analysed scenes, EEG measurements showed that, in women, memory encoding processes were activated to a greater extent than in men (for all scenes). The survey data, however, do not give a clear answer on the subject of remembering an advertisement. When asked whether they had seen an advertisement for a brand, product or service and mentioning various names (including hospice), men recalled the advertisement to a greater extent than women. In contrast, when asked about the details of the advertisement, slightly more women recalled some parts of it. For scenes S2 and S6, half of the women and men were able to recall some parts of it and, for S7, not-significantly more women were able to do so. That can be explained by the fact that not all physiological processes are under volitional control, i.e., enter the conscious sphere [119]. This has been confirmed by the fact that measurements of human physiological processes in specific research situations expand the range of observable parameters and human reactions. Thus, they give us additional information that cannot be obtained from humans by means of declarative methods. This can be explained by the functioning of their subconsciousness. When using declarative methods, it is important how the question is formulated. In the case of closed questions (with answers), it is easier to reconstruct what one has seen before. In open-ended questions (suggesting nothing, hinting) it is more difficult. When the ad was shown again (at the beginning), most respondents recalled its details. It is therefore difficult to draw clear conclusions from such survey research. This applies both to the study of the level of recall by the entire research group and the breakdown by gender. Perhaps an interview would be a more reliable means of declarative research. The non-declarative character of physiological measurements means that they are not based on conscious, expressed human responses to the questions posed to them. Therefore, they are treated as more objective and can be an important complement to declarative research. When studying the impact of advertising on viewers (studying the impact they experience when viewing it), measures based on physiological measurements appear more reliable [24,25].

## 6. Conclusions

Various organizations operating within the health care system use the advertising message to achieve specific goals, usually corresponding with the profile of their operations. Hospices, being institutions functioning within this system, often make use of such forms of communication to raise funds for their activities. The use of cognitive neuroscience methods in this area can improve the effectiveness of such a message. The study of physiological (as well as neurophysiological) parameters of a human body that determine the response to the presented stimuli can significantly broaden the scope of information that can be obtained about the reception of a given message. It provides information that cannot be collected in a course of declarative research.

Nonetheless, studies have shown that physiological measurements (even those performed to a limited extent) provide valuable additional information and are an excellent complement to traditional testing methods. They make it possible to assess and predict behavioral involvement at the population level as regards the reception of an advertising message. Moreover, they help identify scenes (or even single frames), which are likely to affect audiences to a varying degree, i.e., depending on their sex, which is particularly interesting for this study. Due to sexual dimorphism of the human brain, this type of research should be designed accordingly. The results confirmed that different scenes can trigger different cognitive reactions in women and men. This is a valuable indication for designers of advertising messages. The study also showed that statements are not always consistent with neural measurements. The verbal message is the result of processing primary information produced in the brain. It can therefore be subject to intentional or unintentional error. In addition, there are situations in which people are unable to extract information from their memory at a given moment and convey it verbally. It concerns especially feelings, emotions connected with various external stimuli. Additionally, this is exactly what appears in the dynamically changing advertising message.

The knowledge the physiological measures provide may therefore be applied in the process of creating messages targeted at a specific audience and with the intention of shaping specific behavior. This confirms the wide applicability of cognitive neuroscience methods, also in the area of health promotion and health literacy. In this field, the above methods are considered innovative, and their potential opens up the perspective of their growing attractiveness and wider applicability. They also remain attractive in terms of boosting communication between organizations and their environment, and consequently improving the generally understood management processes regarding the issues that are of key importance to these organizations.

## Figures and Tables

**Figure 1 ijerph-18-05331-f001:**
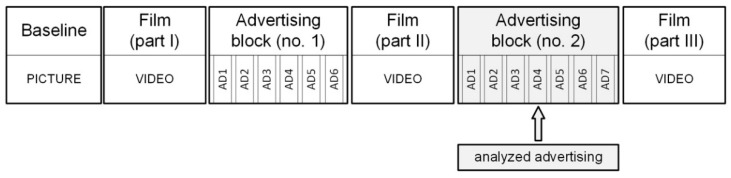
Study protocol structure. The ad under study was placed in the second advertising break.

**Figure 2 ijerph-18-05331-f002:**
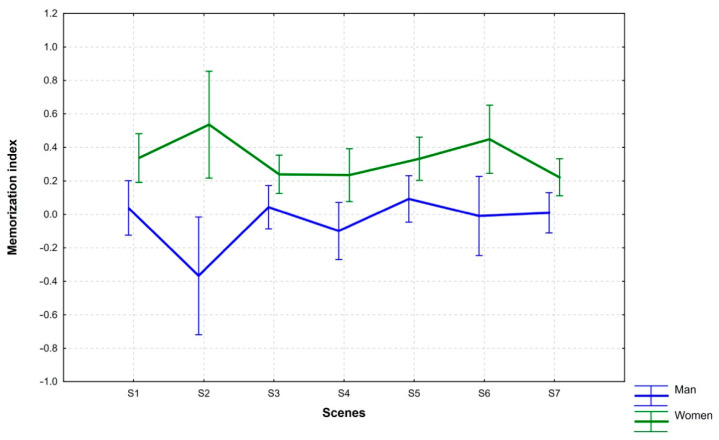
Differences in mean MI values between men and women as regards scenes under study.

**Figure 3 ijerph-18-05331-f003:**
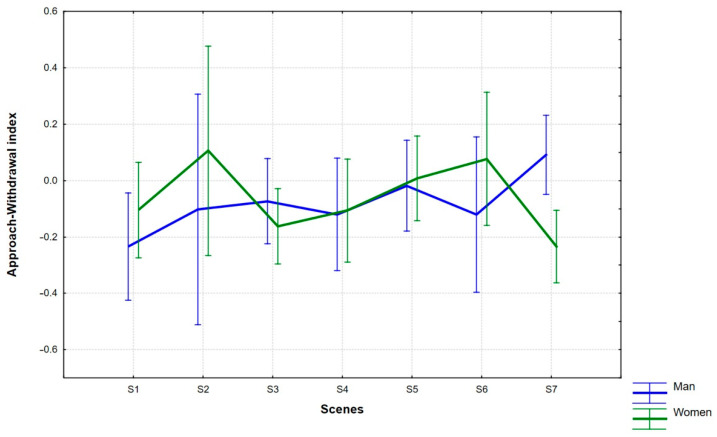
Differences in mean AW index values between men and women for scenes under study.

**Figure 4 ijerph-18-05331-f004:**
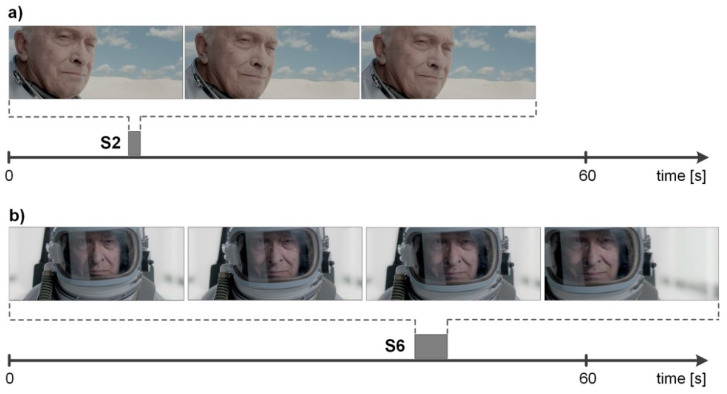
Frames extracted from advertisement under study: (**a**) for S2; (**b**) for S6.

**Figure 5 ijerph-18-05331-f005:**
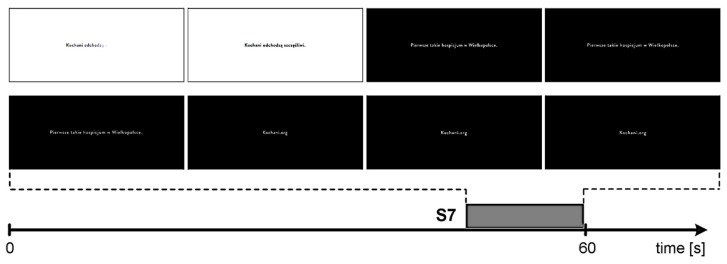
Frames extracted for S7 from advertisement under study.

**Table 1 ijerph-18-05331-t001:** Results of statistical analysis of responses to survey questions given by men and women. Markings: YES (answers: rather YES, definitely YES), NO (answers: rather NO, definitely NO).

Question	Women (%)		Men (%)		N	df	$\chi^{2}$	*p*
	YES	NO	YES	NO				
Do you like the advertisement?	42	10	45	3	31	3	6.590	0.020
Is its message clear?	39	13	39	10	31	3	0.803	0.376 (ni.)
Has it inspired you to donate to support the hospice construction?	32	19	39	10	31	3	2.889	0.268 (ni.)
Do you think that the advertisement will inspire others to donate to support the hospice construction?	32	19	35	13	31	3	0.427	0.519 (ni.)

## Data Availability

The data relative to the study could be obtained by sending an e-mail to mateusz.piwowarski@usz.edu.pl. Dr Piwowarski will return directly the files related to the data gathered by the study.

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
