# Peer review of "Cognitive Neuroscience Methods in Enhancing Health Literacy"

_ijerph, 2021, doi:10.3390/ijerph18105331_

Round 1

Reviewer 1 Report

The study concerns on health promotion, which contributes to health literacy. The purpose of the study tests cognitive responses pressed by neural indices (memorization, interest) to the reception of an advertisement for the 18 construction of a hospice for adults. The findings indicate a different level of interest in the advertisement, which may result from advertising messages concerning health, thus enhancing their performance. It also confirmes the potential and innovativeness of cognitive neuroscience methods in the field. Gereally speaking the study belongs to quantitative study and has empirical significance in  health literacy. The result is slaso enlightening to relevant study.

Author Response

Response: 
We would like to thank you for time that you spent for reviewing our work.

Section of the conclusions was modified and shortened, as suggested.

Reviewer 2 Report

The novelty of the manuscript is its goal to showcase the effectiveness of neuroscience methods in the assessment of social advertisement and message construction using frontal asymmetries in the theta and alpha oscillations as indexes of image and text encoding and retrieval respectively. These implicit measures were followed up by surveys that participants completed after watching the advertisements. The results revealed that men and women elicited differential response to the images and messages with men reacting more to the text and woman more to the images. I found the manuscript compelling in achieving its goal, the exposition quite balanced with providing enough of the big picture and details as well as explaining the advantages and limitations for the reader to form an opinion on the method. I recommend the manuscript for the publication with the International Journal of Environmental Research and Public Health, pending a few minor revisions listed below.

  1. The section 1 Introduction
    1. Needs to be shortened and the main goal of care provider advertisement assessments highlighted sooner instead of waiting to introduce it in the middle of page 2. The language in this section is too verbose with some odd phrasings e.g. p2 line 73-74 “from to the one centered on organizations that provide direct health care” it seems that the referent that would signify the opposite of care providers is missing.
    2. Gender-related differences need to be better motivated in the Introduction. It comes as a surprise to the reader that the main focus of the analyses is the male versus female comparisons. The motivation is briefly highlighted on p14 line 557-562 but it is too late. This bit needs to be moved to the Intro along with any gender-related prior literature on MI and AWI.
  2. Materials & Methods
    1. Have you administered any Handedness tests to determine how many of your participants are right-handed versus left-handed. The hemisphere performance has been tied to handedness as an important factor. If no such tests were administered, please add that as a limitation and a point of investigation for future studies.
    2. EEG set up details need to be added:
      1. what system was used
      2. how many channels
      3. what kind of cap
      4. what was the impedance levels for channels
    3. EEG preprocessing:
      1. Please describe how you dealt with eye movements and other artifacts?This is important especially for your AWI. Did you include the blinks in the analyses or not? Eye movements are the main source of alpha level oscillations in the EEG.
      2. Please spell out what baseline was used on p7 line 341.
  1. Results:
    1. What EEG went into the ANOVA? Averaged over three channels of interest for each participant? Please specify. 

Author Response

Response to Reviewer 2 Comments

We would like to first thank you for your valuable comments and appreciate the time that you spent for reviewing our work. We also admire your vigilance in finding the oversights. We have addressed all your comments as follows and hope you find them satisfactory:

Point 1. The section 1 Introduction:  
a. Needs to be shortened and the main goal of care provider advertisement assessments highlighted sooner instead of waiting to introduce it in the middle of page 2. The language in this section is too verbose with some odd phrasings e.g. p2 line 73-74 “from to the one centered on organizations that provide direct health care” it seems that the referent that would signify the opposite of care providers is missing.

Response:  
The introduction was shortened, the purpose of the article was placed in earlier part of this section. The translation of some phrases has also been improved. The changes are visible in the text.

b. Gender-related differences need to be better motivated in the Introduction. It comes as a surprise to the reader that the main focus of the analyses is the male versus female comparisons. The motivation is briefly highlighted on p14 line 557-562 but it is too late. This bit needs to be moved to the Intro along with any gender-related prior literature on MI and AWI.

Response: 
Changes to the text have been made (from line 117 to line 125), as suggested.

Point 2. Materials & Methods: 

a. Have you administered any Handedness tests to determine how many of your participants are right-handed versus left-handed. The hemisphere performance has been tied to handedness as an important factor. If no such tests were administered, please add that as a limitation and a point of investigation for future studies.

Response: 
An explanation have been added (from line 202 to line 203), as suggested.

b.EEG set up details need to be added:
1. what system was used
Response: 
Details are written in the lines (262-263 and 268-269).

2. how many channels
Response: 
Details are written in the lines (263 to 266).

3. what kind of cap
Response: 
Details are written in the lines (266 to 267).

4. what was the impedance levels for channels
Response: 
Details are written in the lines (267 to 268).

c. EEG preprocessing:
1. Please describe how you dealt with eye movements and other artifacts? This is important especially for your AWI. Did you include the blinks in the analyses or not? Eye movements are the main source of alpha level oscillations in the EEG.
Response: 
The procedure of removing artifacts:
- EEG signal with large artifacts was rejected (3 respondents)
- elimination of the slow-variable trend and high-frequency interference of e.g. muscle origin. - the fifth-order Butterworth filter [2-30 Hz]
- bad channels they were removed (max. one channel) and then interpolated - for 2 respondents
- artifacts from electrodes (short fragments) was flagged and excluded from further analysis.
- eye artifacts were extracted using the ICA method (Infomax algorithm) - only one of the identified components was removed
- next, the continuous signal was divided into one-second time windows (Fast Fourier Transformation) and interference analysis was performed in such intervals
- the following criteria for artifact determination were adopted: amplitude exceeding V and slope of the trend between windows exceeding V/s, with R-squared=0.3. Epochs marked as artifacts were removed from further analyses.

Details are written in the lines (270 to 286).

2. Please spell out what baseline was used on p7 line 341.
Response:
More explanation about baseline have been added (from line 324 to line 327), as suggested.

Point 3. Results: 
a. What EEG went into the ANOVA? Averaged over three channels of interest for each participant? Please specify. 
Response: 
An explanation have been added (from line 329 to line 332), as suggested.

Reviewer 3 Report

In this paper, Piwowarski et al recorded EEG on subjects while they watched a video that included a 60s commercial. After subjects were asked whether they remembered the commercial and what they thought about it.  

The whole paper is well written! The introduction is very well written though also quite long. It reads more like a literature review than an introduction to an article. The conclusion is also long. I leave it to the editor to decide whether these sections should be shortened. I want to emphasize that the whole paper, including the introduction and the conclusion, is very well written – I enjoyed reading it.

My main concern regards the presentation of the results. The authors did EEG recordings and then identified when, during the presentation of a commercial, the activity in frontal regions changed in a way that previous research correlates with improved memory – the "memorization value". Subsequently, the authors also performed a brief survey asking whether subjects remembered the commercial. This survey indicated that the men more often remembered the commercial – thus contradicting the EEG results. The authors mention briefly in the discussion that there are discrepancies, however, this issue should be brought into the open. Should we trust the EEG results or the responses on the survey? 

The authors should also provide more details in Table 1. For example, how many percent indicated that they liked the advertisement. Right now the table only shows the outcome of the gender comparison. 

In short, the authors should expand their discussion about their results and the discrepancies in them. Currently, the discussion and conclusion are somewhat detached from the results of the study. 

Author Response

We would like to first thank you for your valuable comments and appreciate the time that you spent for reviewing our work. We also admire your vigilance in finding the oversights. We have addressed all your comments as follows and hope you find them satisfactory:

Point 1:  
The whole paper is well written! The introduction is very well written though also quite long. It reads more like a literature review than an introduction to an article. The conclusion is also long. I leave it to the editor to decide whether these sections should be shortened. I want to emphasize that the whole paper, including the introduction and the conclusion, is very well written – I enjoyed reading it.

Response: 
Section of the introduction have been shortened and modified. The changes are visible in the text.
Revised as suggested.

Point 2:  
My main concern regards the presentation of the results. The authors did EEG recordings and then identified when, during the presentation of a commercial, the activity in frontal regions changed in a way that previous research correlates with improved memory – the "memorization value". Subsequently, the authors also performed a brief survey asking whether subjects remembered the commercial. This survey indicated that the men more often remembered the commercial – thus contradicting the EEG results. The authors mention briefly in the discussion that there are discrepancies, however, this issue should be brought into the open. Should we trust the EEG results or the responses on the survey?

Response: 
Thank you very much for pointing out. 
The description of the survey responses has been made more detailed, as the one included in the text was vague and imprecise. More explanations have been added in different places (from line 436 to line 453 and in conclusions section).

Point 3:  
The authors should also provide more details in Table 1. For example, how many percent indicated that they liked the advertisement. Right now the table only shows the outcome of the gender comparison.
Response: 
More details in text (from line 462 to line 470) and Table 1 have been added, as suggested.

Point 4:  
In short, the authors should expand their discussion about their results and the discrepancies in them. Currently, the discussion and conclusion are somewhat detached from the results of the study.

Response: 
Section of the conclusions have been shortened and modified. The changes are visible in the text.
Revised as suggested.